# Peripheral Mitochondrial Dysfunction: A Potential Contributor to the Development of Metabolic Disorders and Alzheimer’s Disease

**DOI:** 10.3390/biology12071019

**Published:** 2023-07-19

**Authors:** Most Arifa Sultana, Raksa Andalib Hia, Oluwatosin Akinsiku, Vijay Hegde

**Affiliations:** Obesity and Metabolic Health Laboratory, Department of Nutritional Sciences, Texas Tech University, Lubbock, TX 79409, USA; arifa.sultana@ttu.edu (M.A.S.); randalib@ttu.edu (R.A.H.); oakinsik@ttu.edu (O.A.)

**Keywords:** Alzheimer’s disease, mitochondrial dysfunctions, metabolic disorders, neurodegeneration

## Abstract

**Simple Summary:**

Alzheimer’s disease (AD) is a progressive disease, where dementia symptoms gradually worsen. The causes of AD are complex and are characterized by changes in the brain that lead to the accumulation of two proteins, amyloid beta and tau, forming structures called plaques and tangles, respectively. It is challenging to identify the mechanisms for the initiation and progression of AD. Therefore, multiple hypotheses have been proposed regarding its origin, suggesting mitochondrial dysfunction or metabolic disorders, or both, playing a role. The association of metabolic diseases such as diabetes with AD has been widely studied and more recently, emerged as a promising strategy for AD prevention through anti-diabetic intervention. Further, oxidative stress and inflammation derived from peripheral mitochondrial dysfunction have also been suggested as alternative contributors of AD pathogenesis; however, the mechanism is still unknown. In this review, we summarize the possible interactions between metabolic disorders, mitochondrial dysfunction and AD, and, accordingly, future therapeutic strategies that could target peripheral mitochondrial impairment to prevent and/or treat AD.

**Abstract:**

Alzheimer’s disease (AD) is a progressive neurodegenerative disease characterized by loss of function and eventual death of neurons in the brain. Multiple studies have highlighted the involvement of mitochondria in the initiation and advancement of neurodegenerative diseases. Mitochondria are essential for ATP generation, bioenergetics processes, the regulation of calcium homeostasis and free radical scavenging. Disrupting any of these processes has been acknowledged as a major contributor to the pathogenesis of common neurodegenerative diseases, especially AD. Several longitudinal studies have demonstrated type 2 diabetes (T2D) as a risk factor for the origin of dementia leading towards AD. Even though emerging research indicates that anti-diabetic intervention is a promising option for AD prevention and therapy, results from clinical trials with anti-diabetic agents have not been effective in AD. Interestingly, defective mitochondrial function has also been reported to contribute towards the onset of metabolic disorders including obesity and T2D. The most prevalent consequences of mitochondrial dysfunction include the generation of inflammatory molecules and reactive oxygen species (ROS), which promote the onset and development of metabolic impairment and neurodegenerative diseases. Current evidence indicates an association of impaired peripheral mitochondrial function with primary AD pathology; however, the mechanisms are still unknown. Therefore, in this review, we discuss if mitochondrial dysfunction-mediated metabolic disorders have a potential connection with AD development, then would addressing peripheral mitochondrial dysfunction have better therapeutic outcomes in preventing metabolic disorder-associated AD pathologies.

## 1. Introduction

Millions of people all over the world are living with neurodegenerative diseases. The loss of functionality and the subsequent death of neurons in the central or peripheral nervous system are the hallmarks of neurodegeneration. Yet, the causes for the majority of neurodegenerative disorders are still not well understood; however, the primary risk factor is aging and currently, there are no established cures to slow the progression of neurodegenerative diseases [1,2]. Nevertheless, a few medications may help mitigate some of the mental or bodily distress that comes with these conditions. The two most prevalent forms of neurodegenerative diseases are Parkinson’s disease and AD. With the elderly population increasing in numbers, researchers have estimated that by 2060, 13.9 million Americans will have AD [2].

Evidence from multiple studies has highlighted the involvement of mitochondria in the development and progression of neurodegeneration [3,4,5]. Mitochondria are essential for eukaryotic life since they dynamically generate ATP which contributes to many cellular functions including biological processes, the regulation of calcium homeostasis, the alteration of reduction–oxidation potential of cells, the elimination of free radicals, and the initiation of programmed cell death [6,7]. Disrupting any of these processes affects all cells, but neurons are more significantly affected. Defective mitochondrial function has been acknowledged as a principal contributor to the pathogenesis of common neurodegenerative diseases, especially AD [8,9,10,11,12,13]. Clinical features such as lowered glucose and brain oxygen metabolism, along with numerous microscopic and molecular findings such as alterations in mitochondrial DNA and morphology, and impairment in the respiratory chain function provide strong evidence for mitochondria having a role in AD [4].

AD is generally considered a disorder of the central nervous system (CNS) characterized by the formation of beta amyloid plaques (Aβ) and neurofibrillary tangles (NFTs)/hyperphosphorylated tau protein (P-tau), causing neurodegeneration based on the amyloid cascade hypothesis [14]. The protein responsible for producing Aβ is the amyloid precursor protein (APP) that can undergo different cleaving pathways depending on the cellular environment. The amyloidogenic cleavage of APP regulated by two different enzymes, β- and γ-secretase, generates Aβ, which is pathogenic in AD [15]. Although the production of Aβ is a part of normal physiology, overproduction of the peptide could be toxic.

The protein tau is usually present on the surface of microtubules. This protein provides mechanical strength to the structure of the microtubules, which subsequently facilitates the structural organization of the neurons. However, growing evidence from current studies, especially some unsuccessful clinical trials targeting Aβ production in the brain have suggested the need for the identification of alternative pathways for AD pathology [16]. In the physiological environment, the brain maintains crosstalk with peripheral organs to perform appropriate energy homeostasis [17]. The brain requires a significantly high amount of energy produced by the mitochondria, which mostly use glucose for providing energy to the brain and are capable of protecting the brain against potential stresses. Therefore, any alterations in the mitochondrial metabolism regarding energy supply could have a direct impact on brain function [18,19,20]. Additionally, Aβ can impair the structural and functional stability of mitochondria, promoting AD pathogenesis.

Recent evidence revealed that the underlying pathological alterations in AD, commonly observed in the CNS, are also prominent in the periphery. As stated earlier, mitochondria perform a vast number of cellular functions including energy homeostasis, signal transduction, regulation of stem cells, and providing cellular plasticity for normal functionality under stresses. Dysfunctions in any of the structural and functional features of mitochondria in regions other than the CNS can be referred to as peripheral mitochondrial dysfunction. Patients with mild cognitive impairment (MCI) have been found to demonstrate elevated peripheral inflammation, which is reduced during the severe stages of AD. As inflammatory responses, cytokines and chemokines are released, facilitating the production of ROS and lipid peroxidation. Studies with transgenic AD mice have found increased hepatic and serum TNF-α and IL-6 levels initially prior to locating them in the brain [21]. Pathological alterations that occurred in the AD brain have also been observed in the periphery, indicating a potential association between central and peripheral physiological mechanisms in AD pathogenesis. Moreover, studies have revealed that the impaired function of peripheral mitochondria is a strong indication of primary AD pathology, though the mechanism is still unknown [22].

Metabolic malfunctions, occurring mostly in the peripheral regions of the body, play an important role in the development and progression of AD; therefore, peripheral mitochondrial dysfunction may also have a strong association in CNS degeneration and related diseases such as AD.

Emerging data have suggested that alterations in energy metabolism as well as other structural and functional features of the mitochondria, both in the CNS and the periphery, could likely be a major cause of AD development. This review summarizes the relevant research studies to indicate the potential roles of peripheral mitochondrial dysfunction in relation to metabolic disorders and their association in the development and progression of AD pathologies.

## 2. Current Proposed Hypotheses Underlying AD Pathology

Growing evidence regarding the potential causes and mechanisms of AD pathogenesis has suggested a wide number of hypotheses. As a complex and multifactorial disease, the development and progression of AD appear to involve several mechanisms.

### 2.1. Beta-Amyloid (Aβ)

Under an abnormal physiological environment, the amyloid precursor protein (APP) undergoes cleavage in the amyloidogenic pathway and generates amyloid plaques outside of the cells. These extracellular plaques are potentially toxic to the neurons, perturbing multiple cellular functions, thus triggering inflammation, and mitochondrial dysfunction in the brain [23].

Moreover, studies with animal models and human brains have demonstrated that disruptions in SUMOylation equilibrium can lead to Aβ aggregation and impaired clearance resulting in the onset of AD. Increased expression and moderation of SUMO1 protein with reduced Aβ and APP have been observed in AD mice [24]. In addition, enhanced SUMOylation has been shown to decrease glutamate release in the AD mouse synaptosome while restoration of altered glutamate production can be induced by reduced SUMOylation. Therefore, synaptic alterations stimulated by increased SUMOylation at the beginning of AD might be one of the stimulating factors for AD development [25]. Evidence shows that different SUMO proteins can regulate APP amyloidogenesis in different ways and exert significant effects on Aβ aggregation, indicating a potential role of APP SUMOylation in AD onset and development [24]. Moreover, alterations in the SUMO protein may affect the BACE1 (beta-site APP-cleaving enzyme 1) protein, which catalyzes the initial step of APP cleavage during Aβ production [26]. Thus, differential SUMOylation of BACE1 can regulate the level of Aβ generation and thus influence AD pathogenesis.

Apart from SUMOylation, alterations in glycosylation of several AD-related proteins have been reported with an abnormal glycans profile in AD individuals. Studies revealed the function of N-glycosylated APP in mediating Aβ production, where the associated glycans can assist the APP mobilization [27]. Further modification of N-glycosylation can induce mutations in APP, resulting in a higher Aβ42/Aβ40 ratio, thus indicating a potential association between the glycosylation process and AD pathology [28]. Increased production of Aβ has also been manifested by the glycosylation of BACE1 when modified with bisecting N-acetylglucosamine (GlcNAc). BACE1 is a major target of bisecting GlcNAc, which is regulated by the N-acetylglucosaminyltransferase III (GnT-III), a product of the human MGAT3 gene. AD patients demonstrate enhanced GlcNAc on BACE1 promoting oxidative stress for AD pathology [29].

In addition to the Aβ peptides responsible for AD pathology, recent studies have found Aβ related species, which might originate from a secondary cleavage of the Aβ itself. Differentially truncated Aβ fragments have been reported to be toxic to the cellular environment. Sometimes, the newly truncated fragments could be even more aggressive than regular Aβ and might cause the onset and progression of AD [30].

### 2.2. Tau Protein

Tau is a protein associated with neuronal microtubules and is generally regulated by phosphorylation, playing a significant role in the aggregation and stabilization of the neuronal microtubules. Abnormal hyperphosphorylation of the tau protein present in the AD brain can disrupt the assembly of the microtubules and alter the neuronal structure leading to neurodegeneration. Accumulation of extracellular Aβ plaques stimulates tau phosphorylation; the aggregation of which creates neurofibrillary tangles (NFTs) and results in the disruption of normal cellular function. Due to the lack of functional tau protein, the signaling path in the cell becomes impaired eventually leading to neuronal damage [31].

Additionally, hyperphosphorylation of tau can stimulate SUMOylation resulting in reduced tau solubility and degradation while tau SUMOylation promotes its hyperphosphorylation [32]. A study with APP transgenic mice showed SUMOylation of phosphorylated tau accumulation in the Aβ plaques in AD mice model [33].

Glycosylation is another abnormal modification of the tau protein observed in the AD human brain but not in healthy brain. N-glycosylation of tau may disrupt its aggregation while O-glycosylation has been suggested to have protective effects on hyperphosphorylation related to AD pathology [34]. An in vivo study of AD mice model revealed higher tau phosphorylation with decreased O-glycosylation indicating altered tau modification in AD hippocampus [35].

Based on recent evidence, it is manifested that the tau protein, in addition to hyperphosphorylation, can abnormally be cleaved in the AD brain regulated by proteolytic enzymes. Truncation of tau might occur at multiple sites including both the N and C termini, which generate various fragments found in the AD brain [36]. One study has shown that both N and C truncated tau have pathological activities such as self-aggregation, enhanced hyperphosphorylation, and AD O-tau adoption. Moreover, Tau151–391 was reported to be the most pathogenic among all the truncated fragments. Therefore, therapeutic strategies aimed at preventing tauopathies by blocking tau truncation are being considered [37].

### 2.3. Calcium Hypothesis of AD

The calcium hypothesis of AD was first proposed in 1992 [38]. The calcium hypothesis of AD proposes that amyloidogenic pathway activation may modify neuronal Ca^2+^ signaling cascades accountable for cognitive function. The hydrolysis of APP results in two metabolites with potential effects on Ca^2+^ signaling. Firstly, the amyloids that are secreted into the environment could form oligomers that help the endoplasmic reticulum (ER) to take up the Ca^2+^. Raising the luminal level of Ca^2+^ within the ER improves the sensitivity of ryanodine receptors (RYRs), which in turn increases the amount of Ca^2+^ released from internal reserves. Second, the expression of critical signaling components like the RYR may be influenced by the APP intracellular domain. It is hypothesized that the early memory and learning difficulties in AD originate from this alteration of Ca^2+^ signaling. For instance, remodeling of Ca^2+^ signaling may facilitate the development of long-term depression that relies on the activation of the Ca^2+^-dependent protein phosphatase calcineurin, thereby erasing recently acquired memories [39].

### 2.4. Cholinesterase Hypothesis of AD

The cholinergic hypothesis was first suggested over 40 years ago. This hypothesis proposes that malfunctioning or loss of acetylcholine-containing neurons in the brain contributes significantly to cognitive failure associated with age and AD [40]. AChEIs (acetylcholinesterase inhibitors) are found to improve cognitive function in people with AD by inhibiting the breakdown of acetylcholine [38]. AChEIs are the primary medication approved for treating cognitive impairment associated with AD [41].

### 2.5. Inflammation and Oxidative Stress

Central and peripheral mitochondrial dysfunction are responsible for oxidative stress and inflammatory responses resulting in impaired energy metabolism and neuronal damage. Systemic oxidative stress and inflammation triggered by mitochondrial disruptions have been reported to happen before the initiation of pathological manifestations of AD, such as the accumulation of Aβ [42]. In addition, Aβ can induce mitochondrial reactive oxygen species (ROS) production and subsequent inflammation; thus, significantly destroying mitochondrial structures and functions promoting the neurodegeneration that accelerates AD development [19].

### 2.6. Mitochondrial Dysfunction

Multiple studies have reported defective mitochondrial actions in AD, including mitochondrial bioenergetics failure, oxidative stress, mitochondrial DNA damage, impaired mitochondrial dynamics, mitochondrial membrane permeabilization, and Ca^2+^ dysregulation [43]. Mitochondria, designated as the ‘powerhouse of cells’, is the center of energy production principally through the tri-carboxylic acid cycle (TCA) and the electron transport chain (ETC) utilizing glucose, proteins, and fatty acids as energy sources. The metabolites are catabolized to generate acetyl-CoA, which enters the TCA cycle to produce two types of electron donors such as nicotinamide adenine dinucleotide (NADH) and flavin adenine dinucleotide (FADH_2_). NADH and FADH_2_ move through the ETC, the site of oxidative phosphorylation (OXPHOS) to release energy and facilitate the intermembrane proton pump mechanism to create an electrochemical gradient. Eventually, ATP is synthesized by the ATP synthase utilizing the energy provided by the proton gradient. This entire process is responsible for producing a substantial amount of ROS as by-products through multiple redox reactions. Research has shown excess ROS production impaired TCA and ETC in AD leading to oxidative stress, decreased antioxidants, impaired translocase activity, and dysregulated mitochondrial enzymes [44].

In addition, defective ETC and oxidative stress can result in a vicious cycle of bioenergetic failure involving mitochondrial DNA (mtDNA) damage. Due to its proximity to the OXPHOS complexes, the mitochondrial genome is highly exposed to ROS and thus, is more prone to mutation than nuclear DNA. Further, mtDNA contains a smaller number of non-coding regions, which accelerates the potential pathological impact of even a minor mutation in mtDNA. There is growing scientific evidence that mtDNA mutation is responsible for a wide number of human diseases including aging, neurodegenerative diseases, neuromuscular disorders, and metabolic disease [45]. Damage to mtDNA might influence mutations in several mitochondrial regulators close to ETC, promoting impaired energy production [46]. The coordinated involvement of both the nuclear and the mitochondrial genome is required for mitochondrial biogenesis. The principal regulator of this mechanism is attributed to PGC-1α (peroxisome-proliferator-activated receptor γ co-activator-1α), which turns on several transcription factors, including nrf1 (nuclear respiratory factor 1) and nrf2 (nuclear respiratory factor 2). The nrf1/2 can regulate the expression of mitochondrial transcription factor A, which drives mtDNA transcription and replication. Mitochondrial ROS defense mechanisms involve the association of a PGC-1α-mediated nrf1/2 response against oxidative stresses to maintain normal cellular redox parameters [47].

Mitochondria maintain their physiological equilibrium through the process of fission and fusion that controls the mitochondrial dynamic events including physiological growth and clearing damaged or dysfunctional mitochondria. Uncontrolled fission or fusion results in impaired mitochondrial dynamics causing mitochondrial deformation, overproduction of ROS, damaged synaptic viability, which eventually lead to a disturbed mitochondrial energy metabolism. Studies have shown the potential role of such impairments in several metabolic and neuronal disorders like obesity, diabetes, and dementia [45]. In addition, mitochondrial dynamics is controlled by several mitochondrial as well as cytosolic genes. Three membrane GTPases, including MFN1 (Mitofusin 1), MFN2 (Mitofusin 2), and OPA1 (Optic atrophy protein 1) are involved in mitochondrial fusion connecting two mitochondria at the outer and inner membrane interfaces. Mitochondrial fission involves GTPases including DRP1 (Dynamin-Related Protein 1), DNM2 (Dynamin 2), MID49 (Mitochondrial Dynamics Protein 49), MID51 (Mitochondrial Dynamics Protein 51), MFF (Mitochondrial Fission Factor), and FIS1 (Mitochondrial Fission 1 Protein). A subtle balance between mitochondrial fusion and fission is essential for cell survival and optimal function [48]. Disrupted functions of these genes drastically alter mitochondrial dynamics leading to enormous impairment in mitochondrial homeostatic balance [46].

Cells perform a clearing process called autophagy through which cellular organelles stay stable and healthy [49]. Mitophagy is the method performed by mitochondrial-selective autophagy to clean up damaged mitochondria [50]. Mitophagy pathways are initiated and regulated by multiple enzymes, which are essential for the degradation of non-functional mitochondria. Studies have demonstrated that Aβ and tau-induced inhibition of mitophagy-related enzymes results in impaired mitophagy leading to severe AD pathology [51].

Dysfunction in mitochondria occurs (Figure 1) when any of the above identified phenomena and associated mechanisms are disrupted in the cell. Multiple lines of evidence support the findings that impaired electron transport chain complexes have been identified in the brain of AD patients [52,53]. In addition, mitochondrial dysfunction in the CNS is indicated by a drop in neuronal ATP levels in the brain, which is linked to ROS overproduction and suggests that mitochondria may lose functionality to maintain cellular energy homeostasis [54].

The accurate replication and effective preservation of both nuclear DNA and mtDNA are essential for proper respiratory chain function [55]. Mitochondrial dysfunction caused by mtDNA mutations is primarily maternally inherited, apart from large deletions, which are predominantly sporadic [56]. In contrast, mutations in nuclear DNA are hereditary in a Mendelian manner. The preservation and expression of mtDNA largely depend on nuclear DNA. Therefore, mtDNA functions are significantly impacted by the mutations in nuclear DNA [57].

At the initial stage of AD, when cognitive decline starts, a deficiency in the energy metabolism was shown to occur. In addition, when impaired glucose metabolism takes place, disturbance in several metabolic sensors, for example, the energy metabolism regulated by the adenosine monophosphate-activated protein kinase (AMPK) and insulin signaling mediated by the protein kinase B (PKB) in both central and peripheral organs lead to loss of cognitive skills and progression of AD pathology [19].

## 3. Justification for the Mitochondrial Dysfunction Hypothesis in AD Pathogenesis

Since the discovery of the two pathological hallmarks of AD, the extracellular deposition of Aβ plaque and intracellular formation of P-tau protein have been identified as the cause of neurodegeneration of the brain. The identification of inherited mutations on the APP, presenilin 1, and 2 genes accelerate the accumulation of Aβ, leading to the development of the amyloid cascade hypothesis for AD onset. However, these mutated genes, primarily thought of as the leading causes of AD, are responsible for the occurrence of approximately 1–5% of AD in patients. The major portion of AD, >95% of the total AD population, is attributed to the Aβ plaques and P-tau. This form of AD, called sporadic AD, has been reported to be a result of apolipoprotein E4; a significant risk factor, which may accelerate Aβ formation and plaque deposition. Further establishment of the hypothesis came from the findings of multiple preclinical studies with cells and transgenic animal models showing Aβ plaque deposition as a causative pathogenesis of AD [43,58].

However, several recent studies have found that there is no significant relationship between the level or density of Aβ plaque and cognitive deficits. Moreover, the fact that drugs developed based on the amyloid cascade hypothesis are unable to treat the manifestations of dementia despite reducing Aβ further weakens the hypothesis to some extent. Evidence of late shows that individuals that died due to old age had significant Aβ deposition in the brain without showing any AD-associated symptoms during their lifetime. On the other hand, patients with or without symptoms of cognitive loss were reported to suffer from neurodegeneration even without a diagnosis of Aβ deposition [21,59]. One of the most significant findings to refute the amyloid cascade hypothesis was that treatment with a vaccine could remove Aβ plaques in AD patients, although the pathogenesis of the disease kept progressing [60].

In an aging brain, the normal physiological functions of mitochondria are perturbed including abnormal energy homeostasis, excessive ROS production, and mtDNA damage. Most of these abnormalities are due to dysfunctions in mitochondria, the center of ATP production and regulator of energy homeostasis. In addition, mitochondrial malfunctions lead to increased oxidative stress and decreased bioenergetics and thereby stimulate Aβ production. This finding triggers the thought of a mitochondrial cascade hypothesis, which suggests that altered mitochondrial function plays a significant role in the enhancement of Aβ production and plaque deposition [58]. The above evidence reasonably disproves the amyloid cascade hypothesis and indicates an alternative mechanism termed the mitochondrial cascade hypothesis in AD pathogenesis. The hypothesis infers mitochondrial dysfunction as a central cause for Aβ and tau deposition. This hypothesis, however, has been revised in recent years into primary and secondary cascades. Though the primary cascade hypothesis remains the same, the secondary cascade hypothesis theorized that the accumulation of Aβ induces mitochondrial dysfunction as an intermediate step in the disease development pathway [43]. This hypothesis is receiving more consideration and acceptance from the scientific community due to the emerging data from multiple scientific studies.

## 4. Metabolic Disorders and Alzheimer’s Disease

In 2019, the World Health Organization (WHO) recommended better diabetes management to minimize the risk of cognitive decline and dementia in adults as well as proper monitoring of dyslipidemia at midlife for preventing cognition decline and/or dementia [61]. The translocation of the peripheral Aβ pool to the brain is indicative that the neurodegeneration in the AD brain might not primarily cause the manifestations of physical and mental alterations observed during AD, rather suggesting some other effects beyond the brain [62]. Multiple numbers of epidemiological studies have indicated a possible common connection between metabolic disorders and AD.

Impaired glucose and energy metabolism, among the major abnormalities in AD, indicate a firm association between mitochondrial dysfunction and AD pathology initially. Multiple studies have confirmed the linkage between impaired energy metabolism and altered mitochondrial functions in AD by manifesting decreased enzymatic activities that regulate central and peripheral mitochondrial respiration and TCA cycle such as pyruvate dehydrogenase complex, and α-ketoglutarate dehydrogenase complex, complex II, and complex IV [63,64].

Since the insulin signaling pathway is a significant mediator for both AD and diabetes, studies have postulated insulin resistance as a major connection for the regulation of crosstalk between the periphery and brain and as a preliminary step for the relationship between diabetes and AD [65]. As a characteristic of obesity and diabetes, insulin resistance also occurs in the brain of AD patients resulting in compromised synaptic and neuronal plasticity, which are usually ameliorated by insulin [66]. Peripheral hyperinsulinemia has been reported to be linked with brain atrophy and cognitive decline in AD. Some antidiabetic medications have been reported to improve cognition in AD patients by restoring insulin signaling [67]. Moreover, patients with both diabetes and AD have manifested the downregulation of mitochondrial fatty acid oxidizing proteins, emphasizing that mitochondria play a significant role in connecting AD and diabetes [68].

The regulatory activities of insulin associated with nutrient metabolism and controlling hyperglycemia have been studied in terms of related neurodegenerative diseases. Additionally, hyperglycemia and hyperlipidemia have been reported to be the key risk factors for neurodegenerative disorders, including AD [62,69,70]. Hyperglycemia stimulates the formation of advanced glycated end products (AGEs), which bind to the receptor for advanced glycation end products (RAGE), promoting proinflammatory responses. In addition, RAGE has been shown to accelerate Aβ production in proximity with the blood–brain barrier (BBB). Chronic hyperglycemia also triggers NF-κB and facilitates the secretion of proinflammatory cytokines. Elevated free fatty acids (FFA) produced due to impaired lipid metabolism can disrupt the BBB permeability leading to infiltration of FFA in the brain, thus promoting brain damage and decreased cognition. Chronic low-grade inflammation is the manifestation of obesity, which is characterized by the accumulation of inflammatory cells and higher secretion of proinflammatory cytokines such as TNF-α. Elevated levels of inflammation are devastating for intracellular signaling leading to impaired metabolic functions [66]. It was evident that peripheral inflammatory factors raise the level of Aβ in the AD brain [71]. Thus, metabolic disorders exert a significant influence on creating a favorable cellular environment for the initiation of AD pathology.

## 5. Mitochondrial Dysfunction and Metabolic Disorders

Metabolic diseases, which can affect a wide variety of organ systems, are extremely common across all age groups. In age-related metabolic diseases, mitochondrial dysfunction is at the core of the problem [72]. The primary causes of metabolic disorders include defective cell metabolism, which can result from an imbalance between energy production and consumption. The pathophysiology of conditions like T2D, obesity, dyslipidemia, and cardiovascular diseases have all been linked to mitochondrial dysfunction. Mitochondrial structure, function, and their physiology in metabolic disorders have all come a long way in the last decade (Figure 2) [46].

Currently, obesity has emerged as one of the most significant issues affecting people’s health all over the world. It is recognized as a principal risk factor in the development of a wide variety of metabolic disorders. Recent research showed that mitochondrial malfunction causes oxidative stress and systematic inflammation in metabolic syndrome [73]. Abdominal obesity has been linked to poor mitochondrial biogenesis, which causes mitochondrial malfunction, oxidative metabolism, impaired mitochondrial gene expression, and reduced ATP synthesis in mice and humans. Obese mice overproduce ROS in adipose tissue and change NADPH oxidase and antioxidant enzyme activity. These mice had significantly lower mtDNA, respiratory protein, and Tfam gene expressions [46].

It is well established that mitochondria play a significant role in pancreatic insulin secretion, and consequently, mitochondrial dysfunction is linked to insulin resistance in T2D. Tissue-specific as well as organ-specific studies including liver, pancreas, adipose tissue, digestive system, and the nervous system, targeting mitochondrial dysfunction and T2D reveal that disequilibrium between mitochondrial bioenergetic function, ROS production, and apoptosis have an intimate association with T2D [74,75].

Worldwide, cardiovascular disease (CVD) is one of the main causes of death. The heart, a high-energy-demand organ, requires healthy mitochondria to function properly. Mitochondrial malfunction is considered to be at the core of cardiac diseases, specifically when there are disturbances in respiratory chain activity and ATP generation [76].

## 6. Mitochondrial Dysfunction and Alzheimer’s Disease

The majority of mitochondrial defects directly or indirectly contribute to AD pathogenesis. In other words, studies with the AD brain have revealed multiple types of mitochondrial dysfunction in the brain raising the question of whether such impairments are responsible for AD development, or if the occurrence of AD stimulates mitochondrial defects. It has been demonstrated that mitochondria can be easily accessible to Aβ, which interferes with several indigenous proteins resulting in mitochondrial dysfunction. The sources of Aβ in mitochondria might be those produced in Golgi/ER or the APP associated with mitochondria. A more recent view of mitochondrial APP metabolism has suggested the origin of Aβ in mitochondria is locally produced [77]. The mitochondrial production of ROS influences many metabolic processes and is usually controlled by the detoxification capacity of cellular antioxidant mechanisms. However, in the case of stress or sickness, the oxidant–antioxidant mechanism of mitochondria collapses leading to imbalanced energy production by damaging mitochondrial bioenergetic machineries [78]. Although the mutual causative effects of oxidative stress and the defects of mitochondria are still unclear [79], disruption of the balanced mitochondrial function may set off the pathogenesis of multiple disorders including AD (Figure 3). Studies show that mitochondrial ROS production leads to a vicious cycle in which an elevated level of Aβ triggers mitochondrial dysfunction along with more Aβ production eventually resulting in the advancement of sporadic AD [80].

Dysregulation of functional mitochondria is associated with improper neuronal activity, which requires a well-maintained electrochemical gradient. Neuron cells transfer information in the form of electrical potential and regulate internal communication through the release of neurotransmitters in synapses. The action potential of the neurons at the synapses stimulates the movement of Ca^2+^ ions through voltage-gated Ca^2+^ channels maintaining the secretion of neurotransmitters. Impaired energy production by mitochondria might alter the generation of action potential, Ca^2+^ ion concentrations, and the regulation of neurotransmitters in the brain [81].

Evidence has indicated the association of both central and peripheral mitochondrial dysfunction in the development and progression of AD.

### 6.1. Central Mitochondrial Dysfunction

Mitochondrial defects in the brain, so far, have been reported to be more closely related to AD than those of the peripheries. In transgenic mice, overexpression of mitochondrial Aβ-binding enzyme (ABAD) interacts with increased Aβ in the mitochondria, accelerating neuronal oxidative stress, impaired cognitive function, and memory loss [82]. A study with an APP transgenic mice model demonstrated that the association between APP and mitochondrial membrane obstructed the mitochondrial channels through APP/Aβ deposition on the membrane. As a result, the entry of functional molecules and enzymes was blocked, which in turn led to the accumulation of toxic substances, e.g., hydrogen peroxide [83]. Apart from the APP/Aβ-mediated mitochondrial disorders, the abnormal formation of Aβ is stimulated due to mitochondrial dysfunctions. A similar association of phosphorylated tau with AD neuropathology was shown in human studies using platelet-derived mitochondria demonstrating enzymatic mitochondrial dysfunction in AD progression [84]. Research by Yao et al. with 3xTg female mice demonstrated mitochondrial dysfunction in early stages, even before the obvious accumulation of Aβ plaques or tau tangles had occurred. Altered bioenergetic pathways were shown in the hippocampal cells derived from mouse embryos including reduced mitochondrial respiration as well as enhanced glycolysis [85]. This remarkable study indicated the occurrence of brain mitochondrial dysfunction at an early stage of AD pathology. From a hereditary point of view, the APOE allele is a major genetic risk factor for the initiation of sporadic AD. The impact of APOE on the accumulation of Aβ is well documented with significant evidence of Aβ clearance by APOEε2 gene therapy in the APP Tg2576 mice model [86]. Moreover, further studies with human brains from young individuals having APOEε4, showed reduced mitochondrial COX activity in the brain [87].

In the AD brain, the lipid composition of the mitochondrial cell membrane is altered resulting in a reallocation of proteins associated with the β-amyloidogenic pathway of lipid rafts [88]. A polyunsaturated ω-3 fatty acid, docosahexaenoic acid (DHA), was reported to pass through the BBB and contribute to the construction of the neuronal membrane. This prominent DHA could further accelerate the nonamyloidogenic processing of APP as well as reduce amyloidogenic activities of β- and γ-secretase [89]. Research has shown an association of toxic Aβ production and tau hyperphosphorylation with reduced FoxO activity in the brain under an insulin resistance state. This decreased expression of FoxO results from direct phosphorylation by AMPK when there is insufficient energy available [90,91]. This evidence indicates central mitochondrial dysfunction precedes AD development and progression and influences the central AD pathogenesis.

### 6.2. Peripheral Mitochondrial Dysfunction

Apart from the effects of central mitochondrial dysfunction in AD, the contribution of peripheral mitochondrial defects has also been suggested by multiple lines of emerging data. Research with peripheral tissues such as fibroblasts in humans and mice [92] as well as platelets and lymphocytes demonstrated the association between mitochondrial dysfunction and altered ROS production in the periphery similar to that observed during AD in the brain. Aβ load in the brain is exacerbated through enhanced lipoprotein release in the periphery. The brain maintains its crosstalk with the periphery by lipoprotein ApoE and ApoJ with greater activity of ApoE-Aβ conjugation, resulting in reduced brain efflux of Aβ, which is accelerated when there is lower lipidation of ApoE [93]. Although lipids do not provide energy to the brain, they are significantly present in the mature brain, constituting the membrane structure and performing signal transduction. At the beginning of AD pathogenesis, fatty acids and acetyl-CoA produced through lipolysis and mitochondrial β-oxidation in a fed state participate in ketogenesis induced by inflammatory factors. Under energy deficit conditions, acetoacetate and β-Hydroxybutyrate (β-HB) ketone bodies serve as alternative energy sources for neurons. Generation of β-HB from acetyl-CoA, used as a substrate of acetoacetate, is also stimulated by interleukin-6 (IL-6)-induced p38/NF-κB. The entire process from lipolysis to β-HB takes place in the liver from where the β-HB crosses the BBB to provide the brain with energy under hypometabolic conditions in AD [20]. Moreover, the liver is stimulated by glucose metabolic dysfunction in the brain and supplies ketone bodies to the brain as an energy source in the early stages of AD [94], indicating a connection between the peripheral organ and the central part of the body under adverse conditions.

While the brain obtains energy only from glucose, the peripheral parts can utilize both glucose and lipids as energy sources. Insulin resistance and hyperglycemia in the peripheries can stimulate a bioenergy deficit in the brain, which further deteriorates AD pathology. Central and peripheral alterations in extracellular glucose sensitize the mitochondria and result in AMPK/Akt signal fluctuations by inducing mitochondrial changes. Furthermore, studies have reported a PGC-1α-mediated reduction in mitochondria and PI3K-regulated insulin resistance in both the peripheries and the brain due to the disrupted AMPK pathway. The resulting abnormal glucose metabolism and damaged neurons can be insightful to the connection between hyperglycemia and cognitive decline [95,96].

## 7. Role of Peripheral Mitochondrial Dysfunctions in the Initiation of AD

It is well documented that APP is expressed both in neuronal cells and in peripheral tissues such as the heart, liver, pancreas, blood cells, and kidneys [95]. Moreover, healthy individuals have lower levels of Aβ in the brain and peripheral tissues than that of cognitively impaired individuals. Though there are differences in central and peripheral APP processing and Aβ isoform deposition, there is reduced Aβ in the periphery [97,98,99] and there is evidence of Aβ interconnection between the brain and the periphery. Aβ originated in the brain can be cleared from the brain through several pathways; one of which is the transportation or the efflux of Aβ across the BBB to the peripheral circulation. Apart from the efflux, there could be an influx of Aβ from the periphery to the brain. The potential role of the efflux and the influx might balance the Aβ pool between the brain and periphery. The peripheral mitochondrial origin of Aβ may have systemic effects and stimulate CNS damage after crossing BBB. Several studies including humans and mice have demonstrated the occurrence of Aβ accumulation in the brain when induced peripherally with Aβ extracted from the brain, suggesting its systemic circulation between the periphery and the brain [62]. Studies with APP transgenic mice to test the effects of intraperitoneal injections of Aβ containing brain extract have shown robust cerebral beta-amyloidosis in all the inoculated mice [100]. Further, human studies with subcutaneous Aβ inoculation revealed seeding effects of Aβ, which can spread to the CNS after injection and seed in the parenchyma of the brain [101]. The interaction of Aβ with RAGE in human brain endothelial cells facilitates the transportation of Aβ across the BBB, resulting in the release of proinflammatory cytokines in the brain. Higher expression of RAGE was also reported in the AD brain compared with age-matched controls [102]. These data legitimately suggest the transfer of Aβ produced in the periphery to the brain with a high potentiality of triggering clinical manifestations of AD.

Oxidative stress due to mitochondrial dysfunction has been proposed as a common pathological mechanism underlying most of the chronic neurodegenerative diseases including AD. Whereas inflammation is a body’s protective response against multiple insults, uncontrolled inflammation can result in enormous cell and tissue damage. Due to its vulnerability, the BBB can be highly affected by peripheral inflammation. Under diseased conditions, peripheral immune cells such as macrophages cross the BBB, altering the CNS environment and accelerating chronic neurodegeneration. Further, microglia and astrocytes become activated upon disruptions in the CNS triggering the secretion of proinflammatory cytokines. Moreover, ROS production by activated microglia and astrocytes can be responsible for neuronal damage [103]. Studies with APP transgenic mice have shown increased BBB permeability following peripheral injection of inflammatory substances such as lipopolysaccharide (LPS) facilitating the accumulation of peripheral proinflammatory molecules including TNF-α, and IL-6. Such infiltration of cytokines stimulates neuronal inflammation and subsequent disease progression [71]. Therefore, the evidence of the systemic circulation of Aβ and the impacts of peripheral oxidative stress as well as inflammation on CNS upon crossing the BBB provides potential involvement of the consequences of peripheral mitochondrial dysfunction in the onset and progression of AD.

Based on the above discussion, we summarize (Figure 4) the possible linkages between metabolic disorders, mitochondrial dysfunction, and AD for visualization. There is evidence that peripheral metabolic impairments can influence peripheral mitochondrial dysfunctions. On the other hand, mitochondrial dysfunctions can be the consequence of altered metabolic functions. Therefore, the interrelationship between metabolic disorders, mitochondrial dysfunction, and AD might involve multiple complex and likely common pathways.

## 8. Discussion

Over a decade has passed with no new therapies being approved for the treatment or prevention of AD [104]. Many single-agent medicines have failed to show significant improvement above placebo in research trials. Given the multifaceted nature of AD pathology, combination therapies may be more effective than monotherapy. Several studies have demonstrated that therapies specifically targeting Aβ or phosphorylated tau do not work as anticipated. Researchers’ attention has been drawn to neuroinflammation and oxidative stress as they are considered major players in the development of numerous distinct disorders.

As neuroinflammation is thought to link hypertension and AD [105], drugs such as statins, anti-hypertensives, or their combinations have been investigated and found to reduce the risk of AD in the older population; however, these drugs cannot treat or cure AD [106]. Oral anti-diabetic medications such as thiazolidinediones and sulfonylureas enhanced memory and cognition in AD-afflicted T2D patients [107]. However, these drugs can only treat the symptoms and the mechanisms are yet to be discovered. GLP-1 receptor agonists such as liraglutide were also found to improve cognition in mice models [108] but human studies with liraglutide showed moderate effects in protecting neurons and no effects on Aβ deposition in the brain [109]. Several studies have been performed on metformin and its effect on AD, but the findings are inconclusive. Also, metformin lowers the risk of cognitive impairment in the elderly population. [110]. In another study, after receiving metformin for a prolonged period of time, patients with T2D showed an elevated risk of cognitive impairment [111], which makes it difficult to reach firm conclusions. A case-control study involving individuals with diabetes highlighted that long-term use of metformin was associated with a slight increase in AD risk, whereas long-term use of sulfonylureas, thiazolidinedione, or insulin was not [112].

As the treatments of metabolic disorders did not show the success anticipated, researchers are now focusing on potential underlying causes that may lead to the development of metabolic disorders. Mitochondrial dysfunction has been demonstrated with contributing factors of metabolic disorders including obesity and T2D. In addition, impaired mitochondrial function both in the periphery and brain collectively appears to contribute towards the development of AD pathogenesis. Hence, it is quite reasonable to suggest that mitochondrial dysfunction is a major risk factor influencing the strong association between metabolic disorders and the onset of AD. Therefore, recent studies are focusing on mitochondria-targeted treatment options for preventing the occurrence of metabolic disorders as well as neurodegenerative diseases. Partial inhibition of MCI (mitochondrial complex I) using a tricyclic pyrone compound (CP2) activated several neuroprotective pathways. An in vivo study demonstrated that chronic therapy of CP2 improved mitochondrial and synaptic function, reduced inflammation, decreased Aβ and phosphorylated Tau accumulation, and reduced neurodegeneration in a symptomatic APP/PS1 mouse model of AD [113]. Berberine has been studied for its effects on neuronal health, with positive results in a mouse model of AD, suggesting that it has neuroprotective features [114]. Since berberine indirectly inhibits mitochondrial respiratory complex-I and activates AMPK [115], it might assist in activating PGC-1α and contribute to the biogenesis of mitochondria. In a recent phase-II clinical trial, a combined metabolic activator (CMA) was administered to AD patients and imaging investigations showed hippocampus volume and cortical thickness changes in AD patients that improved cognition. CMA reduces oxidative stress by increasing mitochondrial fatty acid absorption from the cytosol and mitochondrial fatty acid oxidation [116].

For most individuals, insulin resistance is regarded to be the primary cause of metabolic syndrome, diabetes, and vascular disorders like hypertension and cardiovascular disease [95]. Insulin resistance can be initiated by chronic low-grade inflammation and oxidative stress due to the overproduction of ROS in the periphery [117]. Oxidative stress and inflammation also stimulate the development of neurodegeneration. The major player in this scenario is essentially peripheral mitochondria, conferring it as the ultimate target for developing new treatment options for AD.

Based on present evidence, we hypothesize that treatment of metabolic disorders with anti-diabetic drugs in combination with anti-inflammatory drugs, antioxidants, or drugs that target mitochondria may show improvement in AD (Figure 5). Combination drugs or triple therapy may outweigh the failure of monotherapy in the treatment of AD. Further studies are needed to establish this hypothesis.

## 9. Conclusions

Peripheral mitochondrial dysfunctions can induce impaired metabolic functions, or the disruption of metabolic functions can result in mitochondrial alterations in the peripheral regions. Either way, the major consequences are impaired generation and accumulation of Aβ, inflammations, and overproduction of ROS. These consequences may coexist in a cellular environment and influence one another. Metabolic disorders in association with mitochondrial dysfunctions can stimulate the onset and progression of AD. Moreover, the coexistence of mitochondrial dysfunction, metabolic disorders, and AD could indicate a trilateral correlation between metabolic disorders, peripheral mitochondrial dysfunctions, and AD. Despite having several common pathophysiological features, treatments of metabolic disorders have not shown success in improving AD so far. This might be due to the unaddressed damage to the peripheral mitochondria also being a potential underlying risk factor. Restoring and sustaining mitochondrial health may prolong neuronal survival in AD. Addressing the oxidative stress as well as the systematic inflammation caused by peripheral mitochondrial dysfunction in addition to the treatments provided for metabolic disorders may help with better management of the clinical pathophysiology of AD patients.

## Figures and Tables

**Figure 1 biology-12-01019-f001:**
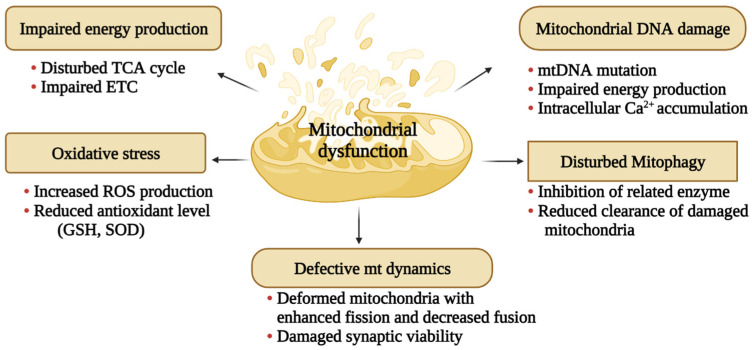
Attributes of mitochondrial dysfunction. Mitochondrial dysfunction characterized by reduced ATP production, bioenergetic failure, impaired dynamics, oxidative stress, membrane permeabilization, calcium ion dysregulation, DNA damage, and uncontrolled fission or fusion leading to metabolic disorders as well as neurodegenerative diseases.

**Figure 2 biology-12-01019-f002:**
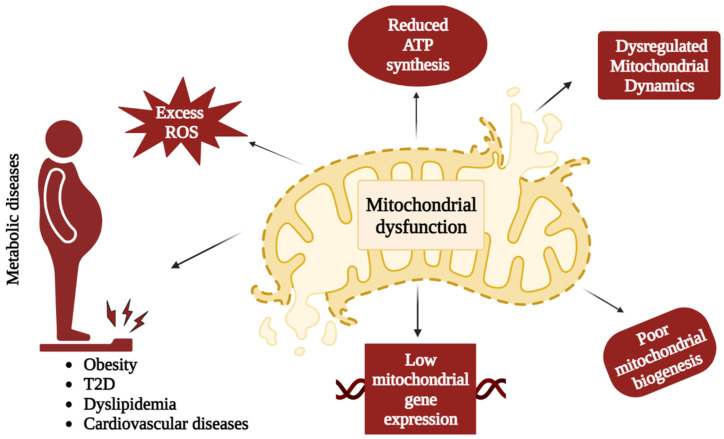
Mitochondrial dysfunction and metabolic disorders. Overproduction of ROS, reduced ATP synthesis, dysregulated mitochondrial dynamics, poor mitochondrial biogenesis, and impaired mitochondrial gene expression may cause mitochondrial dysfunction resulting in imbalance between energy generation and consumption leading to the development of metabolic disorders (obesity, T2D, dyslipidemia, cardiovascular diseases etc.).

**Figure 3 biology-12-01019-f003:**
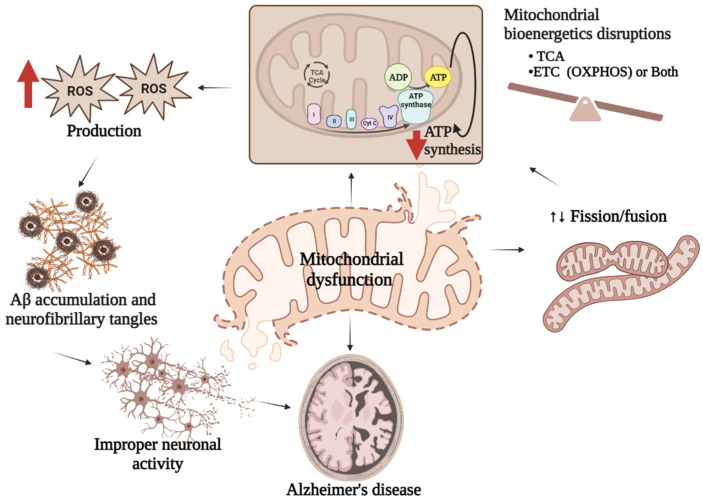
Mitochondrial dysfunction and AD. Mitochondrial dysfunctions characterized either by imbalanced mitochondrial bioenergetics, or impaired mitochondrial biogenesis or both result in reduced ATP synthesis followed by excess ROS generation which stimulate Aβ production leading to improper neuronal activity associated with AD pathologies.

**Figure 4 biology-12-01019-f004:**
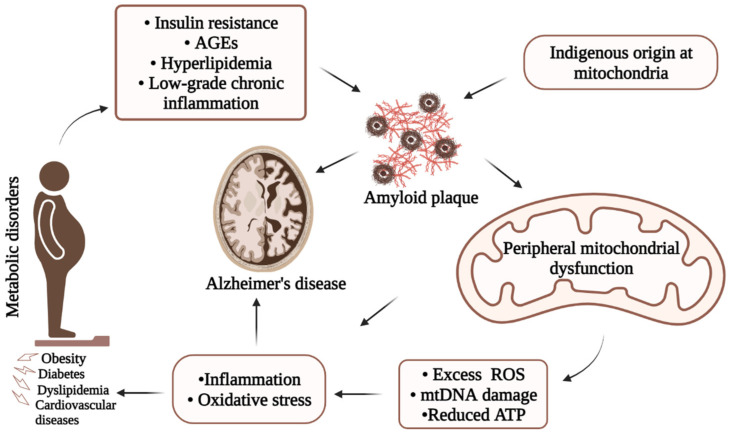
Simplified illustration of underlying AD pathology showing possible interrelation among metabolic disorders, mitochondrial dysfunction, and Alzheimer’s disease. Due to its complexity, onset of AD involves several major pathways including metabolic disorders and mitochondrial dysfunctions through the production of Aβ, increased inflammatory responses, and/or oxidative stresses.

**Figure 5 biology-12-01019-f005:**
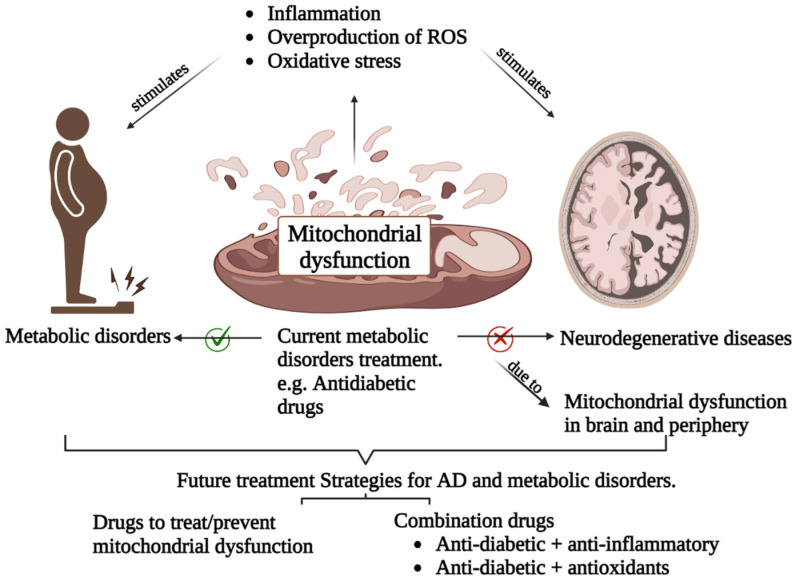
Potential treatment strategies for AD. As an underlying cause, addressing mitochondrial dysfunction may have positive impacts on lowering AD risks either directly or indirectly by minimizing metabolic disorders. Moreover, treatments of metabolic diseases with a combination of multiple drugs might remarkably improve AD.

## Data Availability

Not applicable.

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
