# Peer review of "Peripheral Mitochondrial Dysfunction: A Potential Contributor to the Development of Metabolic Disorders and Alzheimer’s Disease"

_biology, 2023, doi:10.3390/biology12071019_

Round 1

Reviewer 1 Report

Dear authors,

I was invited to review your review entitled “Role of peripheral mitochondrial disfunction in Alzheimer’s Disease: link between peripheral mitocondrial disfunction, metabolic disorders, and alzheimer’s Disease” (Munuscript number: biology-2446079).

This article is focused on the mitochondrial dysfunction, a key player in the development of many metabolic disorders, such as type 2 diabetes, a possible link between these pathologies and Alzheimer’s disease.

Althought the review is well organized, I think that the paragraphs are not explained in a exhaustive manner and some concepts are not clear or not exactly right.

For example, you report that degeneration in AD depends on amyloidogenic hypothesis characterized by the formation of amyloid beta plaques (Aβ) and neurofibrillary tangles (NFTs) of hyperphosphorylated tau protein. But this is only one of the hypotheses proposed and hyperphosphorylation is only one of post-translational modifications of tau protein that triggers neurodegeneration in AD. Today we know that also glycosylation, sumoylation and truncation (with the generation of toxic tau fragments) are important in oxidative stress mechanisms, impairment of synaptic plasticity and mitochondrial dysfunction ect characterized AD neurodegeneration. Furthermore the current literature proposes tau protein alterations as triggering event of amyloidogenesis.

The bibliography just counts 81 references. In my opinion it would be more richer, updated, completed and with more references related to original articles refered to experimental data, in vitro, in vivo (animal models), from human patients that support the thesis on which the autors can build the discussion and conclusions, the numeration is wrong so the citation does not correspond to the reported concept. Furthermore the quote is not always relevant.

The title is not clear, please substitute with another without repetitions.

The english language is good but needs some corrections.

For these reasons I think that the manuscript needs to be rewritten according to all these suggestions.

The english language is good but needs some corrections.

Author Response

  1. In the revised manuscript chapter 2 discusses the current proposed hypotheses underlying AD pathology.
  2. We have expanded the bibliography with a broader range of references that provide a solid foundation for our discussion and conclusions. We have also ensured that the citations are correctly numbered and relevant to the concepts discussed.
  3. The revised title better captures the essence of our review paper and eliminates any unnecessary repetitions.

Please see attached revised manuscript with all the corrections.

Reviewer 2 Report

In this manuscript, the authors reviewed the link between peripheral mitochondrial dysfunction, metabolic disorders, and Alzheimer's disease (AD). The mechanisms underlying this association are complex and multifactorial, but involve insulin resistance, inflammation, oxidative stress, and impaired glucose metabolism. However, I think the authors did not explain the "peripheral" mitochondrial dysfunction clearly. Since this is an important core part of this study, the authors should devote some significant space to exploring and illustrating this point. In addition, I also believe that the authors should strengthen certain discussions, including whether there are potential treatments or interventions that target peripheral mitochondrial dysfunction as a way to prevent or slow the progression of AD. These specific comments are listed below.

1. The part of the abstract only explains the motivation for writing this manuscript but does not excerpt the main findings or discussion of the authors. I suggest that some revisions be made so that the reader can effectively grasp the main points in the abstract.

2. Although the authors have conducted many discussions on the relationship between mitochondrial dysfunction and metabolic disorders, it seems that the relationship between "peripheral" mitochondrial dysfunction and AD cannot be clearly explained. I suggest that the authors should distinguish a subsection to explain why "peripheral" mitochondrial dysfunction can induce AD in mechanism, and present relevant literature or evidence in this section. Since this should be the core of this manuscript, the authors are obliged to enrich the content of this part as much as possible.

3. Although the authors provide many good quality drawings, the lack of explanations of molecular mechanisms detracts from the value of these drawings. I suggest that the discussion of genes related to maintaining mitochondrial function, such as mitochondrial biogenesis or antioxidant capacity, should be added to the discussion, which may be more helpful in developing new ideas in this area.

4. Furthermore, it is recommended that the authors consider adding perspective suggestions in the last part, including possible therapeutic implications and future considerations based on their arguments. This will expand the influence of this review.

English is fine. Some minor typo errors need to be corrected.

Author Response

  1. The revised abstract provides a concise summary of our key points with a clearer overview of the manuscript's focus and contributions.
  2. We have added a subsection to discuss why ‘peripheral’ mitochondrial dysfunction can induce AD. We have incorporated relevant literature and evidence to support our arguments and further emphasize the significance of this core aspect in our manuscript.
  3. To address this, we have expanded the discussion section to include a detailed examination of genes related to maintaining the mitochondrial function, such as those involved in mitochondrial biogenesis and antioxidant capacity.

Reviewer 3 Report

The Manuscript: „Role of peripheral mitochondrial dysfunction in Alzheimer’s Disease: Link Between Peripheral Mitochondrial Dysfunction, Metabolic Disorders, and Alzheimer’s Disease’’ by Most Arifa Sultana and colleagues intends to indicate potential roles of peripheral mitochondrial dysfunction in relation to metabolic disorders in the development and progression of AD pathologies through summarizing relevant research studies. The role of peripheral mitochondrial dysfunction in Alzheimer's Disease (AD) is an area of active research and has gained increasing attention in recent years. In AD, mitochondrial dysfunction has been observed in both the central nervous system (CNS) and peripheral tissues. While CNS mitochondrial dysfunction has long been recognized as a hallmark of AD, emerging evidence suggests that peripheral mitochondrial dysfunction may contribute to the pathogenesis and progression of the disease.

The submitted manuscript, based on already published literatures, attempts to analyse the links between peripheral mitochondrial dysfunction and AD. The manuscript is nicely written with adequate research of literature. After going through the manuscript, I have a couple of comments for the authors:

1.     Studies have pointed out that peripheral mitochondrial dysfunction and metabolic disorders can trigger chronic low-grade inflammation, known as meta-inflammation. Moreover, inflammatory mediators can cross the blood-brain barrier and promote neuroinflammation, which is a prominent feature of AD. Please briefly discuss this point with reference to known studies.

2.     The various proposed mechanisms underlying the relationship between peripheral mitochondrial dysfunction, metabolic disorders, and AD should be discussed in the discussion section.

English is fine. Few grammatical corrections and syntax adjustments are required.

Author Response

  1. In the revised manuscript, we have included a discussion section that briefly summarizes relevant studies indicating that peripheral mitochondrial dysfunction and metabolic disorders can trigger meta-inflammation.
  2. We have addressed the proposed mechanism underlying the relationship between mitochondrial dysfunction, metabolic disorders, and AD.

Reviewer 4 Report

My suggestions:

1. In the figures, I would use a bigger font size, since it may be difficult to read.

2. Mutations in mitochondria may impact several diseases. Authors may add a chapter about this. 

3. It would be nice to mention some neurodegenerative disease genes, which could impact mitochondrial functions (no need to be part of mtDNA). I would also mention whether mitochondrial dysfunction could be either inherited or somatic variants. 

4. Figure 3 is a little too simple. I would summarize it a little more in detail (. a nice example in PMID: 32471464, PMID: 34063708, PMID: 32471464 )

5. I would add a few examples of drug candidates, which target mitochondria and could be promising, even though they were tested in in vitro and in vivo studies.  

Author Response

  1. We have increased the font size in all figures to ensure better readability for readers.
  2. We have addressed how mutations in mitochondria impact several diseases in our revised manuscript.
  3. We have added a subsection that discusses neurodegenerative disease genes affecting mitochondrial functions in the discussion.
  4. We have provided a more detailed summary to figure 3 to ensure that readers can better grasp the essential information presented in the figure.
  5. We have expanded the "discussion" section to include several examples of promising drug candidates tested in in vitro and in vivo studies.

Round 2

Reviewer 1 Report

The authors are properly answered to the questions of referee.

Reviewer 2 Report

The authors have responded to all questions I raised. I suggest the editor may consider accepting this manuscript to be published in its current state.

Reviewer 4 Report

The authors fulfilled my suggestions, thank you.